# Eco-Evolutionary Dynamics in Microbial Communities from Spontaneous Fermented Foods

**DOI:** 10.3390/ijerph181910093

**Published:** 2021-09-26

**Authors:** Anna Y. Alekseeva, Anneloes E. Groenenboom, Eddy J. Smid, Sijmen E. Schoustra

**Affiliations:** 1Laboratory of Genetics, Wageningen University and Research, 6700 HB Wageningen, The Netherlands; anneloes.groenenboom@gmail.com (A.E.G.); sijmen.schoustra@wur.nl (S.E.S.); 2Laboratory of Food Microbiology, Wageningen University and Research, 6700 HB Wageningen, The Netherlands; eddy.smid@wur.nl; 3Department of Food Science and Nutrition, School of Agricultural Sciences, University of Zambia, Lusaka 10101, Zambia

**Keywords:** microbial community, food fermentation, model system, experimental evolution, eco-evolutionary dynamics

## Abstract

Eco-evolutionary forces are the key drivers of ecosystem biodiversity dynamics. This resulted in a large body of theory, which has partially been experimentally tested by mimicking evolutionary processes in the laboratory. In the first part of this perspective, we outline what model systems are used for experimental testing of eco-evolutionary processes, ranging from simple microbial combinations and, more recently, to complex natural communities. Microbial communities of spontaneous fermented foods are a promising model system to study eco-evolutionary dynamics. They combine the complexity of a natural community with extensive knowledge about community members and the ease of manipulating the system in a laboratory setup. Due to rapidly developing sequencing techniques and meta-omics approaches incorporating data in building ecosystem models, the diversity in these communities can be analysed with relative ease while hypotheses developed in simple systems can be tested. Here, we highlight several eco-evolutionary questions that are addressed using microbial communities from fermented foods. These questions relate to analysing species frequencies in space and time, the diversity-stability relationship, niche space and community coalescence. We provide several hypotheses of the influence of these factors on community evolution specifying the experimental setup of studies where microbial communities of spontaneous fermented food are used.

## 1. Introduction and Scope

Ecosystems are composed of various interacting species that in combination determine the functionality of the community. While through theory and comparative studies many concepts of ecological and evolutionary processes shaping these ecosystems have been developed, experimental tests of these have been lagging. The reason for that was a lack of suitable experimental model systems and affordable sequencing technology to analyse the species composition of these communities in many replicates and samples simultaneously.

Stochasticity and various selective pressures constantly affect ecosystems in terms of their biodiversity and functioning. Patterns of biodiversity can be described by species richness and proportions of individuals over these species. Due to selection pressures from the environment these patterns can change, analogous to selection on populations with standing genetic variation, in a process called species sorting [1,2,3]. In the longer term, novel mutations in specific species may be fixed within the community due to selective processes. In this way, changes in biodiversity patterns due to environmental selection can be regarded as an evolutionary process and referred to as eco-evolutionary dynamics [4].

Studies on how evolutionary forces shape and maintain this diversity have mainly been comparative and retroactive: reconstructing the path of evolution by observing and interpreting current (evolved) states of communities. Evolutionary research on a complex natural community is challenging due to a large diversity of organisms on different trophic levels and their interactions. Thus, most experimental evolutionary research is performed using highly simplified systems, mainly (micro-)organisms representing one or a few genotypes with a short generation time. In many fields of biology, the concept that “everything is connected” is extensively discussed and studied, resulting in models of metabolic networks, genetic regulatory networks and trophic structures. Experiments using complete communities from a natural environment, such as spontaneously fermented foods, could provide insights in interactions and their long-term eco-evolutionary dynamics occurring in nature.

Currently, evolutionary research rarely links with ecology by including experiments addressing the consequences for ecological dynamics let alone the evolutionary impact of this connectivity, mainly due to technical difficulties. However, understanding the influence of evolution on co-existing organisms could deepen our ecological understanding of community performance and allow us to manipulate community function, e.g., nutrient fluxes and test community for resilience and stability.

To advance, natural model systems are required with a limited number of species and interactions. A few of these have been reported, such as microbial communities in tree holes, self-assembled and synthetic communities derived from marine environments [5,6,7,8]. Traditional fermented foods, containing mixed communities of fermenting microbes, add to such natural tractable systems comprising of microorganisms, making them suitable for experimental ecological and evolutionary research [9,10].

From this perspective, we elaborate on the types of ecological and evolutionary questions that may be experimentally addressed using microbial communities as model systems. We show various experimental models that have been applied to study communities in terms of their ecology and evolution and how these can add to our understanding of community performance by showing what theory can be tested. We also discuss methodological approaches to study those model systems. In Appendix A, Appendix B and Appendix C we provide additional information about experimental evolution as well as a short outlook on possibilities for research applications.

## 2. Microbial Model Systems

Already in the 19th century Dallinger started controlled evolution experiments with microbes—or at least invisible organisms (Appendix A). A large-scale follow-up of this new way of studying evolution came much later [11,12,13,14]. Microorganisms are of interest to evolutionary biologists because they are small, have short reproduction times and can easily be stored and preserved for long periods of time [15]. The short generation time of microorganisms allows us to see evolution in action and even attempt to predict evolution [16]. The ability to store bacterial cultures by freezing and later thawing them without loss of viability allows for direct comparison and even competition experiments between evolved and ancestral types [15].

Due to these advantages microorganisms already mitigated a lot of challenges in evolutionary research by allowing experimental approaches to complement comparative studies such as the use of phylogeny to study evolution [17]. In this section we explain various model systems of increasing complexity that are used for evolution experiments.

### 2.1. Single Strains

Most experiments focus on the evolution of a single bacterial strain in a defined laboratory environment [18,19], allowing for direct tests of specific variables on the outcome of evolutionary processes. This includes the dynamics of adaptation to novel environments and stress conditions such as the presence of antibiotics, patterns of repeatability of evolution, predictability of evolution through the mapping of evolutionary pathways on adaptive landscapes, elucidation of trade-offs in evolution and evolutionary constraints at the genome level [13,15,20,21,22,23,24].

As most microorganisms live in nature in close proximity to hundreds or even thousands of other bacterial species and organisms from other taxa, the approach of experimental evolution using microbes could be expanded to the community level. However, analyses of most natural environments would result in massive quantities of data, which might make the formulation of predictions for evolutionary experiments difficult. As a solution and to simplify these microbial communities, they can be shaped into synthetic communities with only a limited number of focal (micro-)organisms.

### 2.2. Synthetic Communities

Naturally co-occurring microorganisms can be isolated from their environment and brought together in the lab in pre-determined concentrations. These so-called synthetic communities can be used for studying evolutionary processes under strictly defined conditions [25,26,27,28,29,30]. Due to previous interactions and co-evolution, these bacteria could represent the essential parts of a natural community compared to combinations of laboratory strains which have no historical connection [31,32]. In this way, synthetic communities are assumed to represent nature more accurately than most artificial communities, where species/strains are not evolutionary adapted to each other, while keeping the simplicity that is needed for experiments. A very elegant synthetic community was constructed recently by combining 33 strains from a range of environmental sources [33]. Upon long-term propagation through serial transfer of 48 days in a laboratory environment, around half of the strains were maintained in the community that could be analysed for a range of functional properties [33].

Using synthetic communities also poses several challenges. First, researchers sometimes struggle to isolate the key species from the community. Some community members might occur unculturable and will therefore be excluded from the community of isolates. Other bacteria that were isolated might not have been a member of the natural community but were incidentally present. The second challenge is to achieve the relevant degree of diversity. A very simple model will not represent nature accurately. Lowly abundant species may be still important for ecosystem functionality but may be missed when assembling the synthetic community. In an experiment using a microbial community from cheese, an initially very rare species became predominant upon long-term propagation in a novel environment [34]. Further experiments showed the influence of bacteriophages on culture diversity. Researchers found that the simple bacterial blends used in their experiments did not represent the diversity of the original complex starter culture enough to evaluate the role of phage predation [35]. Later, Spus repeated the experiment with more complex blends of bacterial strains and highlighted the impact of phage predation on community diversity [34].

### 2.3. Self-Assembled Communities

Another approach is to look at how communities assemble and stabilise in composition. In 2017 Friedman and his colleagues showed that the community assembly can be predicted: in pair-wise competitions, species which can co-exist in pairs will survive in more complex community. Eight species were successfully assembled together in a stable community in well-controlled laboratory conditions based on prior pairwise co-culturing tests [28]. However, such predictions could (partially) fail due to factors such as variation in initial proportion of the community members [36] or stochastic frequency oscillations [37]. Stochastic niche theory postulates that new species establish within a community only if their offspring can survive stochastic mortality while growing to maturity on the resources left unconsumed by earlier established species. This theory was supported by a series of simulation experiments [38]. Further experiments of Goldford and his collaborators showed the self-assembly of stable complex communities in simple growth media. Bacteria for those experiments were taken from soil and leaf surfaces and were phylogenetically diverse. The established community converged reproducibly to reflect the experimentally imposed conditions rather than the initial inoculum [39]. Current research indicates that the stabilised by self-assembly microbial communities consist of metabolic generalists, rather than metabolic specialists. However, few studies on disentangling the exact functional relationship in a complex community have been done recently [40].

Although the approach of extracting strains from natural communities into synthetic is very valuable, these two challenges might make the translation into “real world” ecology and evolution unrealistic [41]. This motivated the search for a better model system. What we need to find are small confined “islands” of microorganisms in which the number of players and their interactions is limited and therefore manageable. In these “islands” no selection or extraction of species into synthetic communities is required for communities to be experimentally tractable.

Bell and colleagues found these “islands” in the form of small pools formed by the roots of beech trees [42,43]. Researchers further performed a series of evolutionary experiments with those isolated communities and found out that a stronger evolutionary response occurs in low-diversity communities, and the increase in community diversity or the genome size of the focal species can be linked to a lower adaptation capacity [44]. Another model to study ecological and evolutionary processes is microbial communities from “half-natural” wastewater systems. These contain a few key players which can be identified using NGS sequencing and provide new insights in community assembly and niche mapping. Manipulating those microbes in controlled environment of wastewater systems in time can be a promising option to understand community dynamics and evolution [45].

As the number of players in the pools on beech tree roots or wastewater is limited, it is not necessary to extract players and put them together in a set frequency. These communities can directly be used for experiments. Moreover, those ecosystems can be rebuilt by using isolated strains for experiments with a lower diversity. Consequently, all the steps between single strain behaviour towards the behaviour in natural systems can be compared. The risk of losing vital interactions will be low and observations in the lab could represent nature.

### 2.4. Traditional Fermented Foods with Complex Microbial Communities

Traditionally fermented foods can form another ‘island-group’ of interest [46,47,48,49]. Many traditional fermented foods rely on spontaneous fermentation, which means that they have little human interference as they are not produced using defined starter cultures but are fermented by a naturally available microbial community. These natural communities are usually diverse but not too complex, e.g., up to 13 main players and numerous other species at very low abundance in three traditional fermented products from Zambia (Schoustra et al., 2013).

To improve organoleptic properties of fermented foods, producers often re-use a finished fermented product to start a next batch of the same product [50], in this way propagating the microbial community that underlies fermentation. In the food science domain this process is referred to as back-slopping [51]. Back-slopping can also occur passively, by the re-use of non-sterilised fermentation equipment, like previously used vessels [52]. These vessels will become the long-term habitat of the fermenting microorganisms. In other production methods back-slopping is done actively, like in Illa-type Mabisi, a fermented milk product from Zambia, as well as for parmesan cheese production [53,54]. The so-called natural whey starters which are used to produce Parmigiano Reggiano (parmesan cheese) consist of bacteria which have co-existed for long periods of time with enough nutrients available to go though many generations. This method is intended to make a stable quality product but can be compared with a standard evolutionary experiment (outlined in Appendix B). The diverse microbial interactions are therefore assumed to be more like those of an evolved community.

Many traditional fermented foods are dominated by communities of lactic acid bacteria [55,56,57,58] The physiology, metabolism and genetics of lactic acid bacteria have been elaborately studied because of their dominant role in fermented food communities [59,60,61]. A series of evolutionary experiments with single strains of lactic acid bacteria prove that those are a suitable model object for studying eco-evolutionary processes [62,63]. Other microorganisms such as acetic acid bacteria, yeasts and moulds can also play significant role in fermented foods. In addition, bacterial viruses, bacteriophages, have recently been recognised as crucial members of food microbial communities and taken more into account as also playing a key role in community performance [64]. New phage-filtration and isolation protocols in combination with sequencing and bioinformatic analysis allow us to estimate the role of bacteriophages in food production and food microbiology, e.g., clarify the geographical origin of a product, predict the community composition after several cycles of batch fermentation [65,66]. The extended knowledge of metabolite production and growth profiles of these microorganisms in fermented foods can help to understand the observed evolutionary pathways. The ongoing development of affordable sequencing techniques as well as methods of data analysis now make it feasible to characterise large numbers of communities required for studying the evolutionary outcomes of replicated evolutionary experiments.

There are many fermented foods that rely on spontaneous and uncontrolled fermentation and contain a diverse microbial community. Not all of them are suitable for evolution experiments due to complex production protocols, consistency or composition complexity. In our opinion, fermented liquids, such as milk or beverages, with communities containing bacteria, could be the most convenient concerning experimental set-up and culture or DNA based downstream processing. The natural situation might be even better represented by communities that include other species from different groups such as yeasts, moulds and viruses. This increase in community diversity and complexity, in turn, might also increase the complexity of sequencing and analyses. On the other hand, such diversity brings the model system much closer to natural ecosystems with various trophic levels, niche distribution and stability.

Using food products as a model system stimulates collaboration between fundamental research groups focussing on evolution and research groups working in the field of food sciences and applied microbiology. This multidisciplinary approach is expected to lead to fermented food with improved properties. An example of this type of research can be found in Appendix C.

In summary, the microbial communities present in spontaneous fermented foods make a very useful and interesting model system for evolutionary research, complementing the few other model systems that have been developed [6,43,67,68]. Three aspects contribute to this: (1) their limited complexity that still represents nature, (2) the production methods allow for the communities to adapt to their environment, and (3) the available knowledge about the individual players in the community. In the next section we explain some concepts concerning community evolution that can be addressed using experimental microbial communities.

## 3. The Experimental Study of Eco-Evolutionary Dynamics

The suitability of a model system depends on the experimental design for testing hypotheses on changes in biodiversity patterns due to selective forces. This process can be described as species sorting [69] or as eco-evolutionary dynamics, which can be formally defined as “interactions between ecology and evolution that play out on contemporary time scales, with “contemporary” intended to encompass time scales on the order of one to hundreds of generations” [70]. Consistent changes in patterns of biodiversity can be viewed as an evolutionary process—in line with changes in allele frequency in populations with standing genetic variation in response to selection [3,71].

Various theories concerning eco-evolutionary dynamics of communities have developed [72,73,74,75,76]. Here, we highlight some theories that could be addressed in experiments using microbial communities from natural systems such as fermented foods. For these theories we also indicate what the experimental setup could look like. Figure 1 indicates how much different model systems mentioned in Section 2 score on various aspects on a scale of 1 (low score) to 10 (high score). While the score is somewhat arbitrary, it does provide a graphical representation of a comparison concerned with to what extent four types of model systems would represent nature, how easy it is to study individual species and community structure over time, and it shows how well different evolutionary questions and theories mentioned in this section can be addressed. Table 1 lists (dis)advantages of discussed model systems for answering different evolutionary questions.

### 3.1. Patterns of Diversity

In natural communities, species frequencies vary in space and time. In these communities, variation patterns can be measured and differences in these patterns can be linked to potential causal factors [67,77]. A study focussing on baker’s yeast revealed that analyses of eco-evolutionary dynamics can be done by sampling microbial community in the same food product or the same type of product but derived from different geographical regions and over time. The slight alterations in environmental factors or production methods of those fermented foods diversify the selection pressures that shape the microbial community structure [67]. Diversity richness in microbial communities in fermented foods depends on the level of “industrialisation” of the production process: the more handcrafted the product is—the greater the diversity in it [78]. The differences and similarities found can also be linked to environmental data. This has been inspirational for modelling studies focussing on traits and exploring the way ecological factors, such as nutrient cycles, symbiosis or competition, and natural selection can influence community structure in coevolving species [77,79].

Current modelling approaches are even more complex and manipulate several species as well as their metabolite fluxes, covering inter-species interactions, resource availability, quorum sensing and stochastic processes across community “propagation” [80]. Tackling the complexity of community modelling by splitting community networks into pairwise interactions yields good results: a study on plant interactions with soil within biome gave many insights into forest ecology, nutrient limitations and succession [81]. In addition, a model incorporating the data-driven approach combined with such pairwise community defragmentation allowed to predict community dynamics through time-series of generations in artificial gut microbial community harbouring twelve key species [82].

Communities from milk-fermented foods seem like a suitable platform to create models to generate and verify predictions from an eco-evolutionary perspective: those communities contain a few key species of lactic acid bacteria, yeast and viruses with known patterns of metabolism and inter-species interactions. However, this field of using modelling in food microbiology has just recently opened [83,84]. Taken together, observations of different patterns allow us to generate hypotheses and predictions that are testable using experimental communities.

Testing of predictions on what factors have the biggest influence on community diversification can be done by challenging the microbial communities in the laboratory in a selection experiment. These factors can be related to the degree of diversity of the community, the number of niches that are available in the environment, as well as the evolving interactions within the community. All these aspects apply to microbial communities in spontaneous fermented foods and other natural microbial communities.

### 3.2. Diversity-Stability Hypothesis

The biodiversity–stability hypothesis poses that a more diverse system has greater stability in terms of functionality [85]. The functionality of microbial communities in fermented food products is based on their ability to convert the available nutrients in the food matrix into metabolites to obtain the required product characteristics. The clear definition of functionality allows for an easy assessment of the loss or change of functionality, e.g., unsuccessful acidification, reduced breakdown of proteins or off-flavour production. Greater species diversity within the microbial community can result in a stable functionality of the community due to a back-up function [30,42,86]. If, for whatever reason, certain members of the community are not present anymore, a diverse community might contain members that can take over the lost function. Greater diversity also causes a smaller number of unoccupied niches, by using more nutrients [87]. In that case, occupied niches are not available for any invader, which makes the whole system more likely to keep its functionality and not be destabilised by a non-cooperator, such as a food-spoiling or pathogenic microorganism [26].

Whether a natural community is more stable when it is more diverse can be tested by manipulating these natural communities to become less diverse. During propagation in an evolution experiment, a fraction of the communities is periodically transferred to fresh medium (Appendix B). By using sequential propagation with an extremely high dilution factor of the inoculum, only those bacteria present in the highest numbers will remain, which strongly decreases the diversity in the community. Whether the diversity which is lost was crucial can be tested by studying the change in fermented end-product characteristics and testing the resilience of the microbial community against stress or invaders. Additionally, the influence of diversity on evolution of the community itself can be tested [43].

Sometimes predators can facilitate stability. These non-cooperators can stabilise a diverse community when growth rates of different members greatly differ. The member with the highest growth rate is the preferred victim of a predator according to the “kill the winner” principle [34,88]. The fastest grower in a community could potentially provide most nutrients for a predator and will therefore be the preferred pray, keeping the community stable. This phenomenon is closely related to “negative frequency dependent selection” of the focal strains, where an increase in frequency of an organism has a negative effect on the fitness of that organism [21,89,90].

In context of fermented foods, the role of a force of common species frequency regulator might be bacteriophages and yeast viruses [91]. Although the first steps to characterise the viromes of fermented foods were made already 10 years ago [92], many current studies are still focused mainly on the characterisation of viral communities [93]. In fermented kimchi bacteriophages were persistently found throughout many days of fermentation in a high fraction of mainly Siphoviridae and Myoviridae. These viruses were suspected of controlling the diversity of bacterial community composition by maintaining the “kill the winner” strategy [94]. However, it would be useful to incorporate viruses from microbial community into evolutionary experiments and models of microbial communities as this brings the system closer to natural conditions and provides insights into community stability and functioning.

Apart from predation, negative frequency-dependent selection can also be caused by other limiting forces, like food resources, cross-feeding or physical space. The influence of these forces can be studied by reconstructing natural bacterial communities using frequencies that differ from the frequency found in nature. The speed with which these communities will return to their original frequencies, if they do, can give an indication of the strength of these forces.

### 3.3. Niche Space

Over the years, various hypotheses have been developed concerning niches in established communities. One of the oldest hypotheses concerning niches is the niche exclusion principle which states that one niche can only be occupied by one organism [95,96]. If two species occupy the exact same niche, descendants of the most fit organism will gradually take over from the descendants of the less fit organism. In natural environments the niche is defined by various depletable resources, like nutrients and space, and non-depletable resources, like temperature and pH, which together form a multidimensional niche space [97]. Because of these different dimensions, in theory, organisms can live together in a community as long as one dimension in the niche space is not overlapping between the two organisms [98,99,100]. The magnitude of the overlap of niches determines the level of competition between the species. This allows for many different organisms in a natural community as there are many different niches available.

Free niches might increase the chance of invasion by an alien species [87,101,102]. In practice, we still do not see all the possible niches, with all possible combinations of dimensions that can be occupied. Species that might exist in a community cannot coexist with already established players in the community due to competition for non-substitutable resources. The availability of unoccupied niches may result in character displacement [103] and adaptive radiation [104], whereby organisms adapt to occupy other niches. A classic example of microbial adaptive radiation is the work on *Pseudomonas fluorescens* in a static environment, where initial monocultures diversify into at least three phenotypically different types that each specialised on a specific niche that is defined by an oxygen gradient [104].

Yeast and lactic/acetic acid fermentations have been studied from the perspective of niche specialisation during domestication of the starter cultures. Upon domestication, fermenting microorganisms loose/expand parts of their genomes under applied selection pressure and evolve higher fermentation rates [105,106,107]. Spontaneous fermentation along with a huge variety of traditional production methods created a potential global “bank” of microbial communities with a diversity of selection pressures and niche combinations, creating stable and resilient communities. At the same time, industrialised fermented products usually contain only part of such natural communities, leaving free niches open in the ecosystem [108,109].

Without external fluctuations, the number of niches will remain stable and should be equal to the number of species present in the community. By propagating a microbial community with different numbers of species or by changing niches, it can be tested whether the number of species and the number of niches are indeed so strongly linked [95,96]. This could be tested experimentally with a system where the performance and resilience to withstand perturbations of a full natural community can be compared with the performance of a synthetic community of increasing complexity assembled from species taken from that natural community. Fermented foods seem to be a suitable model system for such experiments.

### 3.4. Fluctuating Environmental Factors

In natural communities we cannot neglect external (environmental) fluctuations. The continuously changing environmental factors in batch culturing are to a certain extent comparable to fluctuations like seasons and tides in natural ecosystems. Under such dynamic conditions, nutrient-rich periods are alternated with nutrient-poor periods. These fluctuations may allow different organisms to flourish in different moments of time, resulting in a more diverse community. There could be a trade-off between growth rate in the exponential phase of fermentation and survival rate in the stationary phase that is following the fermentation which will result in a balance of organisms that have a different strategy [110,111]. While predicted in theory, these microbial experiments have demonstrated that such a trade-off exists [112,113].

The influence of these fluctuations on community structure can be investigated by varying the time regime of the batch fermentations that is also characterised by phases of rapid population expansion followed by periods of stationary phase at the species community level. In this way, the balance between fast growers and those with high survival could change. It is possibly very difficult to completely eliminate some players in the community [34], but it is hypothesised that when there is no stationary phase, the community will mainly consist of fast growers, while at constant low nutrient levels, the community will consist of mainly slow-growing survivors, like it was observed in lactic acid fermentations [114]. Temperature regimes also affect the core microbiome composition and therefore, overall community composition and functional state [115]. Another crucial factor is nutrient availability: nutrient limitation /caloric restriction may lead to fixation of specific strains/microorganisms, such as nitrogen-fixing and cellulose producing bacteria in kombucha fermentation, of which domination might not be beneficial when media is rich in nutrients [116,117]. Thus, community modification can follow several directions of stabilisation under the influence of environmental fluctuations: physiological adaptation and specialisation, compositional shift and evolution of new strains under selection pressure. Additional studies applying fermented foods as model systems on the stabilisation direction (or a combination of directions) upon environmental fluctuations would bring more understanding to the community assembly and stability.

### 3.5. Community Coalescence

The term “community coalescence” was introduced recently [118] and describes situations where entire communities interact because the environments they are in are physically mixed, sparking this interaction. In nature we see this, for example, during soil tillage and flooding events, but also in humans while they are eating or kissing. The performance of single species does not always give a good indication of how the species will behave in a community [119]. The coalesced community might be a combination of the two initial communities or be dominated by either of them, depending on the best performing combination [120].

The influence of co-evolution in the outcome of community coalescence can be investigated using microbial communities from fermented foods. Combining two or more co-evolved communities can provide information on how specialised the evolved interactions within the communities are. In addition to this coalescence of similar communities, also the mixing of a fermented community with the community of a raw product has important implications for understanding the principles of community assembly and resilience as well as the insights on manipulating the fermentation technology. Studying the influence of co-evolution on the outcome of community coalescence, or comparing results of multiple regular coalescence occasions, can be tested in experiments with fermented products.

A further interesting avenue here is the beneficial interaction between microbial communities from fermented foods and the gut microbiota [121,122,123]. The interactions between these communities have presumed profit that warrants further study, for instance by adding spontaneous fermented foods into simulated gut systems and following the gut microbial community dynamics and functionality. Such experiments promote collaboration between specialists from different areas which might be insightful and productive [124,125].

## 4. Conclusions and Outlook

Global concern about maintaining the ecological diversity remains acute. In order to solve ecological issues a strong scientific background should be built [126] for which experimental approaches can complement theoretical and comparative studies. Natural ecosystems are often complex and hard to disentangle, compromising the study of processes driving the composition and functionality of those ecosystems. To advance, defining the role of each member of an ecosystem is crucial in understanding of community diversity, stability and resilience, but it is also a challenge. That is especially true for some microbial communities, like the ones from soil [127], human gut [128] or fermented foods [129] which are diverse, but contain many unculturable species. Recently developed new-generation sequencing techniques with a combination of bioinformatics and modelling allow us to learn about community structure and are important steps [130,131,132]. In recent years the meta-omics approach has become more affordable and allows to systematically disentangle the complex network of interactions between microorganisms also in experimentally evolved communities [133].

Microbial communities in some spontaneously fermented foods provide an opportunity for experimental research since they contain diverse and microbial ecosystems that are relatively simple, but still stable over time [9,54,134,135]. Applying fermented foods as a model system allows to experimentally test evolutionary theory and to follow eco-evolutionary community dynamics, such as the effect of selection pressures on changes in patterns of biodiversity. This would complement other experimental systems that have been developed recently and would bridge work on synthetic and natural communities.

Microbial communities like the ones from spontaneous fermented foods bear several intrinsic advantages for executing evolution experiments: short generation times, small size and ability to be stored frozen and defrosted to perform competition experiments (fitness tests) between evolved and ancestral lines. Moreover, these natural microbial communities have a limited number of players and form an island of microorganisms that does not have a lot of influx from the outside the confined system boundaries. These communities also contain well-studied microorganisms (many full genomes of lactic acid or acetic acid bacteria as well as yeasts are available) with a clear function which makes it easy to follow up on evolutionary changes.

However, using fermented foods as an experimental model also poses certain challenges. First of all, often using natural communities as a model system requires a transfer of such community from natural environment into controlled environment of the laboratory where conditions are different from natural. This may affect community performance and stability. To overcome this, one could plan field studies where community stay as close as possible to natural conditions. Additionally, handling a complex community requires more effort to control the spatial heterogeneity, to decrease risks of possible contamination (in a complex community it is harder to quickly detect an invader). In microbial ecosystems viruses play an important role and it is a challenge to control/follow the community dynamics as many species affected by the virus are divided into susceptible and resistant subpopulations which could be harder to track [136]. There is also demand in increasing the sample replication to cover for natural stochastic variation in composition and functionality of communities. However, when the richness of microbial community is decreasing, deterministic processes may take over natural stochastisity like it was shown in soil communities [137]. Many traditional fermented foods became (half-)industrialised and got adapted to well-controlled conditions [138], creating a gradient from completely natural to completely controlled ecosystems where one could choose the level of control/domestication of community when planning an experiment.

All these characteristics make bacterial communities from fermented food products an interesting model system to test long-standing theories in community ecology and evolution and increase our understanding of evolution and its drivers. For example, an experiment on species depletion through series of dilutions can be performed to test the role of key/less abundant species in the community stability/resilience over time. An evolutionary experiment on both diluted (where only top abundant species are present) and undiluted full community could provide us with an understanding of the extent to which the less abundant species support community stability and whether they can take over the function of one of the key species. Another important question is to study the role of environmental alterations or nutritional components on community stability, composition and function over a series of transfers. One way to test this is to use several types of milk (skim milk, lactose-free milk, goat milk, plant-based milk, etc.) for dairy fermentation and propagate the communities in a series of transfers. Such an experiment could shed light on the environmental impact on the taxonomic and functional diversity of microbial communities and whether we can steer community into desired direction by changing part of the environment, in particular by affecting fat/sugar/protein content in milk environment). Evolutionary experiments on natural and artificially constructed model systems may be considered as a new type of ecosystem engineering—“evolutionary engineering”, where species become adapted to each other and result in more stable fine-tuned community.

## Figures and Tables

**Figure 1 ijerph-18-10093-f001:**
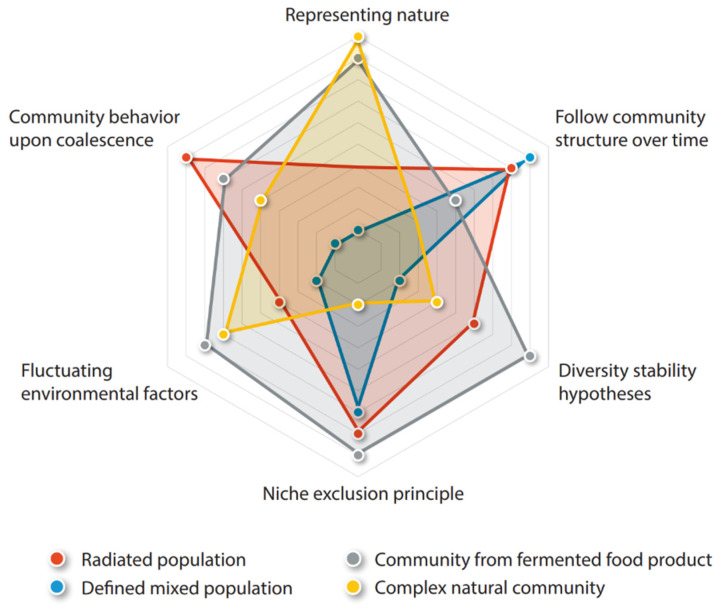
Different model systems and their applicability for experimental study of evolutionary theory. Values are a combined value for representability of nature and the ease of using the model systems in the experimental setup. The lowest score (1) is in the center and the highest score (10) is on the outside of the sphere. For now, more complex communities have the disadvantage of more complicated analyses, where not all players can be isolated and studied individually. When isolation and typing techniques become more available and affordable due to developments in the field of sequencing, the applicability of more complex communities as a model system will increase.

**Table 1 ijerph-18-10093-t001:** Advantages and disadvantages of various model systems. This table lists the advantages and disadvantages of using various model systems mentioned in the paper to analyse different evolutionary hypotheses. The different hypotheses are listed in the top row, followed with a short description in the second row.

	Patterns of Diversity	Diversity Stability Hypotheses	Niche Space	Fluctuating Environmental Factors	Community Coalescence
	Communities can show radiation, be divergent or convergent	More diverse communities are more stable	Number of niches available gives a maximum for the number of stable members in a community	Nutrient-rich and poor periods can make communities more diverse and more flexible	Co-evolved communities will maintain their function upon coalescence with a less stable community
Single genotype that radiated into a mixed population	All players in the community can be followed exactly	Diversity is very low	Very limited number of niches is filled in the community	Small communities have little potential to respond to fluctuations or might be too instable for fluctuations	Origin of remaining players can be easily analysed
Defined mixed population	Communities can be analysed, more complex community structures can be followed	Due to initial instability of the communities the link with diversity can be challenging	Niche space can be constructed in an organised or pre-defined manner	Well-constructed communities might have enough potential to show flexibility and increasing diversity	Origin of remaining players can be analysed well
Community from fermented food	Most players can be analysed, natural behaviour can be followed	Natural communities with different diversities and similar function can be used	Bacteria are fully adapted to their niche. Bacterial interactions make full community analyses challenging	Natural communities contain a lot of microbial potential to see how adaptive they can be towards challenging environments	Challenging to see the origin of the final community structure, however, functional groups can be interchanged
Complex natural community	Groups of players can be followed over time	Most communities found in nature are too diverse and/or too stable to find the relation	Bacterial interactions shape niche space and make analyses of all niches challenging	Natural communities contain a lot of microbial potential to see how adaptive they can be towards challenging environments	Challenging to see the origin of all players to see how the communities behaved after coalescence has taken place

## Data Availability

Data sharing not applicable–no new data generated.

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
