# Peer review of "Eco-Evolutionary Dynamics in Microbial Communities from Spontaneous Fermented Foods"

_ijerph, 2021, doi:10.3390/ijerph181910093_

Round 1

Reviewer 1 Report

I read with interest the review "Eco-evolutionary dynamics in microbial communities from spontaneous fermented foods". The review reports an interesting analysis on the different microbial models proposed so far, analyzes their advantages and disadvantages in their use and concludes that lactic fermentations can constitute an excellent model for ecological-evolutionary studies. Very accurate, well written and with interesting reflections.

Some suggestions:

Figure 1: Redraw the figure more clearly. In particular, the writings are not read well.

Line 306: "enfironmental" correct with "environmental"

Lines 414, 554 and 616: put the name of the bacteria in italics.

In References section: I suggest leaving a space between “doi:” and the reference number, so that you can log in automatically, as in reference 42.

Author Response

Dear Reviewer,

Thank you very much for your time and for a positive feedback on our paper. We value your suggestions and made all corrections that you kindly pointed out: 

  •  Figure 1: Redraw the figure more clearly. In particular, the writings are not read well. We re-drew the figure using Adobe Illustrator and provided a separate file with the figure in high resolution for editors. 
  • Line 306: "enfironmental" correct with "environmental" : corrected 
  • Lines 414, 554 and 616: put the name of the bacteria in italics : corrected 
  • In References section: I suggest leaving a space between “doi:” and the reference number, so that you can log in automatically, as in reference 42 : corrected  

 All corrections were made in "Review mode" in MS Word and a revised document was resubmitted. 

Reviewer 2 Report

The submitted manuscript has presented an overview of various common experimental model systems which are used to study eco-evolutionary processes. Among these model systems, the authors recommend the Microbial communities of spontaneous fermented foods for its complexity of a natural community, extensive knowledge on many of its community members, and ease of experimental setup at lab scale. A couple of examples using the model system in research on eco-evolutionary dynamics were provided. Overall, the manuscript provides insightful perspective on various models used in research on eco-evolutionary forces. However, there are a few revisions needed before it can be published in our journal.

Major revisions

  1. Challenges associated with using microbial community from fermented food are not sufficiently covered.
  2. No specific examples of studies using microbial community from fermented food on Niche Space and fluctuating environmental factors were provided.
  3. Need to clarify the scoring system in more details for different models evaluated in Figure 1.

Minor revisions

  1. On page 8 line 351, “In that case occupied niches” (instead of “In that case accupied…”)
  2. On page 8 line 306, “environmental factors or …” (instead of “enfironmental factors or …”) 

Author Response

Dear Reviewer, 

 We are grateful for your time and for a positive and constructive feedback on our paper. We addressed your comments in major revisions point by point and integrated this feedback into our paper: 

  • Challenges associated with using microbial community from fermented food are not sufficiently covered.  -- We agree that there was a lack of summarizing the challenges of working with fermented foods as a model system and we believe it is important to highlight not only benefits, but also challenging points. We now added a paragraph on this topic in Conclusions and Outlook section and elaborated on how to tackle those challenges. 
  • No specific examples of studies using microbial community from fermented food on Niche Space and fluctuating environmental factors were provided.  -- We agree that there were few examples of such research. We studied literature and added more references indicating that there is background on this topic. However, we indicate that more research in this field is required in order to clarify the processes of niche specialization and the role of environmental fluctuations in composition and functionality of complex ecosystems. 
  • Need to clarify the scoring system in more details for different models evaluated in Figure 1. -- We added clarifications in the text above the figure as well as in the figure caption. We admit that the score is somewhat arbitrary, however it expresses our opinion on using different models applicability for experimental study of evolutionary theory. We formulated this scoring system based on our experience in the field and on revised literature on this topic. 

We also made corrections according to two minor revisions (page 8 lines 306 and 351). All corrections were made in "Review mode" in MS Word and a revised document was resubmitted. 

Round 2

Reviewer 2 Report

The authors have addressed all of my concerns, and I have no objection for the revised manuscript to be accpeted for publication.